# Stereolithographic Additive Manufacturing of Zirconia Electrodes with Dendritic Patterns for Aluminum Smelting

**Masaya Takahashi [1],* and Soshu Kirihara [2]**

1   Graduate School of Engineering, Osaka University, 2-1 Yamadaoka Suita, Osaka 565-0871, Japan
2   Joining and Welding Research Institute, Osaka University, 11-1 Mihogaoka Ibaraki, Osaka 567-0047, Japan; kirihara@jwri.osaka-u.ac.jp
*   Correspondence: masaya-takahashi@jwri.osaka-u.ac.jp

**Abstract:** Zirconia electrodes with dendritic patterns were fabricated by stereolithographic additive manufacturing (STL-AM). A solid electrolyte of yttria-stabilized zirconia (YSZ) was selected for oxygen separation in the molten salt electrolysis of aluminum smelting without carbon dioxide excretion. Thereafter, 4, 6, 8 and 12-coordinated dendritic structures composed of cylindrical lattices were designed as computer graphics. The specific surface area of each structure was maximized by changing the aspect ratio. The spatial profile and surface pressure of the hot liquid propagation in the dendrite patterns were systematically visualized by computational fluid dynamics (CFD). During the fabrication process, a photosensitive resin containing zirconia particles was spread on a substrate, and an ultraviolet (UV) laser beam was scanned to create a two-dimensional (2D) cross-section. Through layer laminations, three-dimensional (3D) objects with dendritic structures were successfully fabricated. The ceramics were obtained through dewaxing and sintering.

**Keywords:** additive manufacturing; ultraviolet laser lithography; yttria stabilized zirconia; aluminum smelting

## 1. Introduction

High-temperature ceramics of yttria ($Y_2O_3$)-doped zirconia ($ZrO_2$) exhibit effective ion conductivities through substitutional solid solutions of low-valence cations [1]. Oxygen ions can be transferred in the solid phases along the oxygen vacancies formed in the zirconia–yttria composite lattice [2]. The cubic crystals are stabilized to restrict thermal transformations in the tetragonal and monoclinic crystals above a molar concentration of 8.0, as shown in the zirconia–yttria phase diagram [3]. The processed ceramic components of the yttria-doped zirconia linearly expand and contract with repetitive heating and cooling [4]. Solid electrolytes of yttria-stabilized zirconia (YSZ) have been developed for practical applications in solid oxide fuel cells (SOFCs) [5] and high-temperature oxygen analyzers [6].

Recently, YSZ electrodes have been used in the molten salt electrolysis of aluminum smelting [7]. Metal anodes of mesh tubes coated with fine YSZ layers can attract oxygen ions in molten alumina, reducing the melting point by cryolite addition. The oxygen ions are absorbed at the surfaces and transferred through the electrolytes, and gaseous oxygen is formed at the mesh anodes and drained through the tube. The remaining aluminum ions are gathered at the cathodes, and the deposited metals can be refined using pure metals. Conventionally, carbon anodes are introduced into molten alumina, and oxygen is chemically separated as carbon dioxide. The use of YSZ electrodes for metal refining is a promising method to reduce greenhouse gases.

In our previous studies, YSZ electrodes with three-dimensional (3D) structures were precisely fabricated by stereolithographic additive manufacturing (STL-AM) for practical applications of SOFCs [8]. Microlattice patterns with ordered pore arrangements were designed and created to increase the specific surface areas for effective gaseous reactions

and permeabilities. Photosensitive resins with ceramic nanoparticles have been used as printing inks in STL-AM [9]. A cross-sectional layer was formed on a spread slurry paste by laser drawing. A solid object was obtained via layer lamination and interlayer bonding in the lump of uncured paste. The formed composite precursor was dewaxed and sintered to create a full ceramic electrode [10].

Here, YSZ microlattices with ordered porosities were processed as ceramic electrodes for effective aluminum smelting without discharging carbon dioxide. The coordination numbers and aspect ratios of the ceramic microlattices were spatially considered and optimized to systematically increase the molten salt permeabilities and specific surface areas by numerical simulations. The laser processing conditions in STL-AM were systematically optimized and progressively discussed. The dimensional accuracies and microstructural soundness of the ceramic lattices were measured and precisely observed after heat treatments.

## 2. Experimental Procedure

The dendritic lattice structures were designed by a computer graphic application (Autodesk: Fusion 360), as shown in Figure 1a–d. The cylindrical lattices were connected at the nodal points with coordination numbers 4, 6, 8 and 12 and spatially propagated at equal angles. The lattice aspect ratios, defined as the numerical proportions of diameter to length, were systematically modified from 0.75 to 3.00 to maximize the specific surface area. The designed unit cells in the cubic envelopes were continuously arranged to create lattice structures with ordered porous patterns. All the cavity paths were continuously connected and diverged along the lattice coordination. The periodically formed nodal points had the same coordination number.

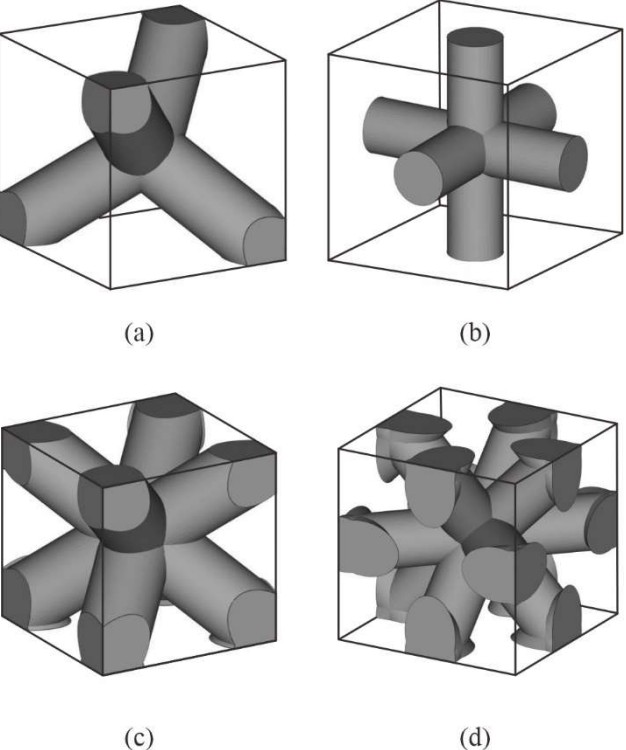

(a)　　　　　　　　　(b)

(c)　　　　　　　　　(d)

**Figure 1.** Graphic models of dendritic lattices in cubic unit cells. Cylindrical lattices were connected at coordination numbers of 4, 6, 8 and 12, as shown in (**a–d**), respectively. These units were continuously copied and parallel-translated in three dimensions to create the lattice propagations.

The liquid phase streamlines were visualized by a computer fluid dynamics (CFD) application (Cybernet: Ansys Workbench). The pressure and temperature distributions at the surfaces and insides of the connected lattices were simulated using the finite element

method (FEM). In the calculation area, the dendritic structures with lattice constants of 3.54 mm were spatially arranged for $2 \times 2 \times 2 = 8$ units, as shown in Figure 2. The molten salt of the aluminum and cryolite mixture was assumed to be a high-temperature fluid at 1000 °C. The density and viscosity were 2.75 g/cm$^3$ and 2.58 mPa·s. The pressures at the inlet and outlet were 1.01 and 1.00 atm. No-slip conditions were applied at the liquid and solid interfaces.

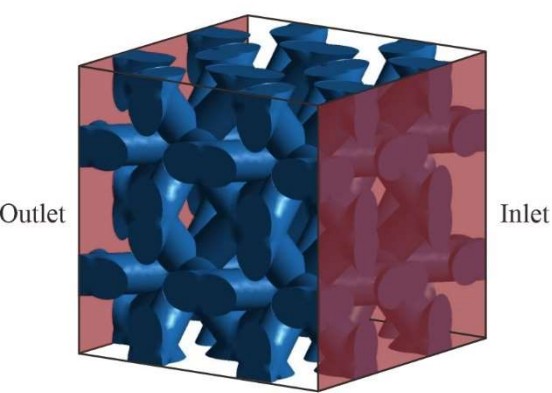

**Figure 2.** A numerical simulation area with the dendritic lattice of $2 \times 2 \times 2 = 8$ units. Streamline distributions of the molten aluminum were visualized by computer fluid dynamics (CFD). Temperature and pressure distributions on the lattice surfaces were estimated using the finite element method (FEM).

YSZ particles (DKK, HSY-0293) with an average diameter of 520 nm were dispersed into the acrylic resin (JSR, KC-1287) at a volume fraction of 45%. The formed paste was sealed in a polyester container with a capacity of 500 cc. The air bubbles were removed by centrifugal forces during revolution movements, and the particles and resin were kneaded in rotation movements using a mechanical mixer (SK-Fine, SK-350T). The rotation and revolution speeds were configured at 700 and 500 rpm. The batch processes with mixing times of 5 min were repeated for three sets. Subsequently, the paste material was packed into a syringe barrel with a capacity of 500 cc, and the syringe unit was mounted onto the STL-AM equipment (SK-Fine, SZ-2500).

The paste material was injected from the syringe unit by pneumatic pressure and spread on a metal substrate using a mechanically movable knife edge. An ultraviolet (UV) laser with a wavelength of 355 nm was focused on a beam with a diameter of 50 μm and scanned on the paste surface at a moving speed of 2000 mm/s. A testing film (edge length of $5.0 \times 5.0$ mm) with a round hole (1.0 mm in diameter) was drawn, as shown in Figure 3. The irradiation power of the laser beam was increased from 50 to 250 mW to optimize the lithography conditions. The film thicknesses and hole diameters were systematically measured according to the irradiation power to estimate the curing depths and part accuracies using a digital optical microscope (DOM) (Keyence, VH-Z100).

The sequential STL-AM process is schematically illustrated in Figure 4. The cross-sections were consecutively formed by the laser drawing through photopolymerization in the spread paste layers. The solid component was successfully fabricated by layer lamination and bonding under optimized lithography conditions. The unsolidified pastes were removed by compressed air blowing and ultrasonic washing. The obtained composite precursors were dewaxed and sintered at 1350 °C for 2 h in air. The relative density was measured using the Archimedes' method. The microstructures of the YSZ lattices were observed using a scanning electron microscope (JEOL, JSM-6060).

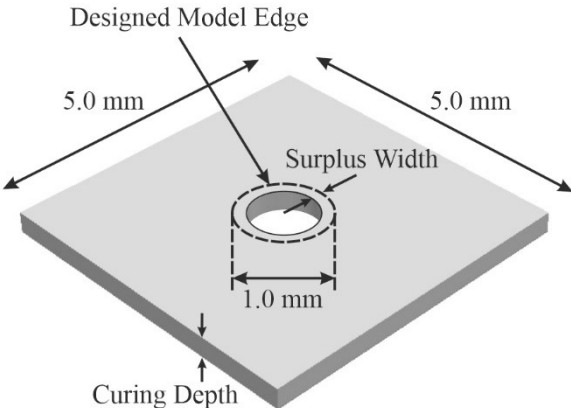

**Figure 3.** A lithographic test pattern to systematically optimize the laser drawing conditions. A round hole (1.0 mm in diameter) was opened at the center of a square sheet (5.0 mm in edge length). The hole diameters and sheet thicknesses were defined as the dimensional tolerances and curing depths, respectively.

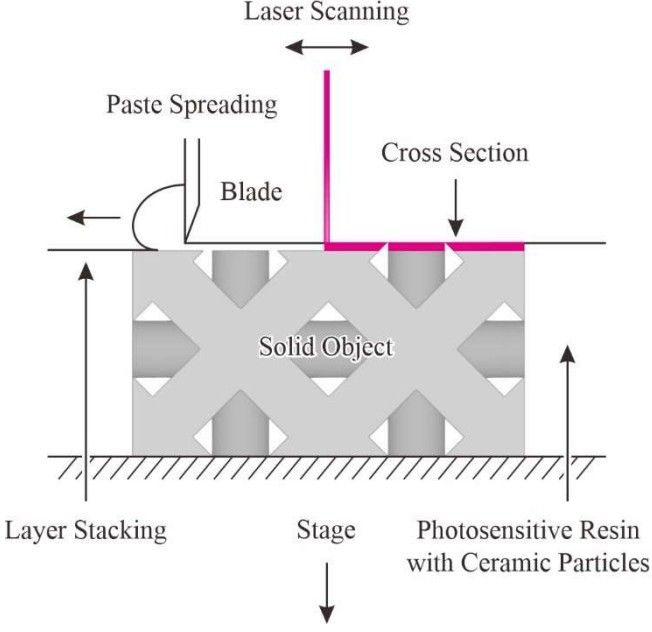

**Figure 4.** A schematic illustration of process sequences in stereolithographic additive manufacturing (STL-AM). An ultraviolet (UV) laser beam was scanned on spread resin pastes with fine particles, to draw cross-sections. A solid component was automatically formed through layer lamination and interlayer bonding.

### 3. Results and Discussion

The specific surface areas of the dendritic lattices were calculated according to the aspect ratios, as shown in Figure 5. These numerical values were obtained under the geometric conditions of a uniform lattice constant. The maximum amounts were individually exhibited at specified aspect ratios of 1.17, 0.90, 2.34 and 2.18 on the lattice patterns with coordination numbers 4, 6, 8 and 12, respectively. These optimized model parameters were used in the structural modeling of the numerical simulations. The specific surface area was maximized in the 12-coordinated lattices, compared to those of the other lattices. However, the fluid flow profiles of molten salts should be considered to optimize the YSZ lattice patterns and increase the separation efficiencies of oxygen ions.

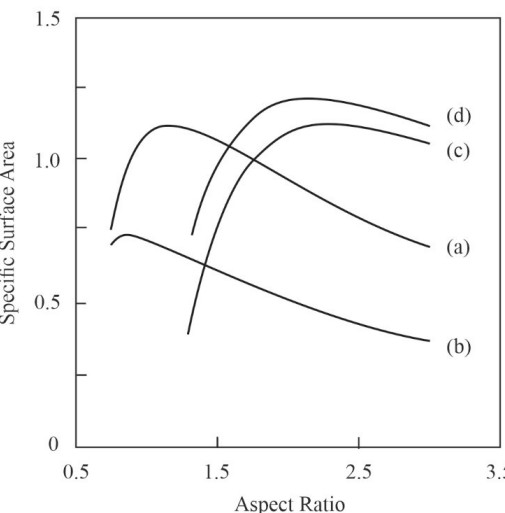

**Figure 5.** Calculated specific surface areas of the modulated dendritic lattices. The aspect ratios were defined as the numerical proportions of the lattice lengths to diameters. The curved solid lines (a–d) indicate the maximum specific surface areas on the 4-, 6-, 8-, and 12-coordinated lattices, respectively.

The streamline distributions of molten salts were visualized to represent fluid movements as graphical contrasts with the YSZ lattice models, as shown in Figure 6. These line densities corresponded with the flow rates, and the color tones indicate the flow velocity. The molten salts were spatially distributed in the 4-coordinated lattices, as shown in Figure 6a. However, the construction points with high streamline densities were formed in the vicinities of the lattice junctions. Straight streamlines were formed in the 6-coordinated lattices, as shown in Figure 6b. The relatively brief contact of the molten salts with the YSZ lattice did not contribute to the oxygen ion absorption at the interface. Smooth meandering streamlines were formed in the 8-coordinated lattices, as shown in Figure 6c. However, retention points with low streamline densities were formed between the nodal lattices. Molten salt streamlines surrounded the 12-coordinated lattices, as shown in Figure 6d. Oxygen ions were absorbed on the YSZ lattice surfaces and conducted the following electric field.

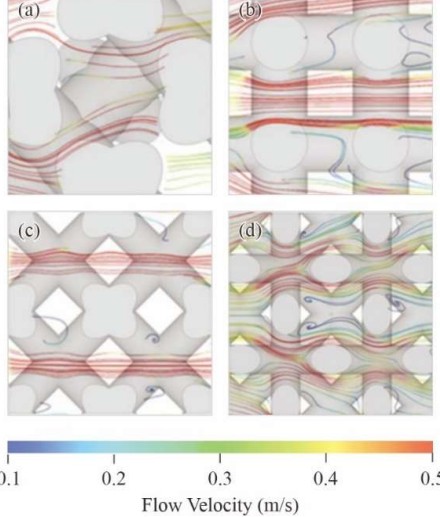

**Figure 6.** Streamline distributions of high-temperature liquids in the dendritic lattices visualized by computer fluid dynamics (CFD). The flux densities and color variations indicate the fluid amounts and velocities, respectively. The fluid profiles (**a–d**) were simulated in the 4-, 6-, 8- and 12-coordinated lattices, respectively.

The 12-coordinated lattices with an aspect ratio of 2.18 were mathematically verified to exhibit the maximum specific surface areas and high liquid permeabilities. The stress distributions loaded from the molten salts for the YSZ lattice surfaces were visualized, as shown in Figure 7. The mechanical stresses were concentrated on the side facing the inlet and vicinity of the nodal points of the cylindrical lattices. The maximum amounts were lower than 400 MPa in the fracture stress of the sintered YSZ at temperatures above 1000 °C [11]. The temperature distributions were visualized in the cross-section parallel to the fluid flow direction, as shown in Figure 8. The lattice temperature gradually decreased away from the electrode surfaces by thermal radiation inside the electrode. The oxygen ion conductivities in the YSZ should be effectively activated during aluminum smelting within the range of 750–1000 °C [12].

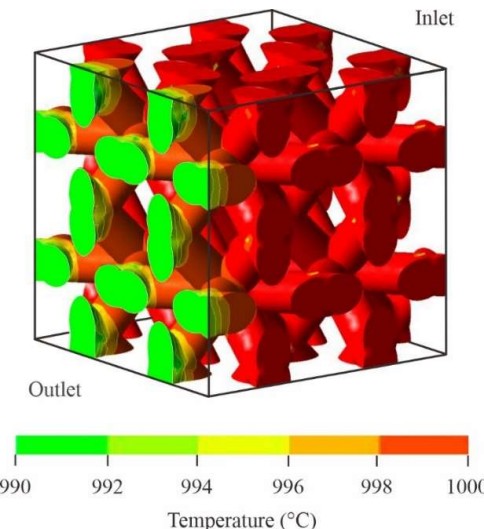

**Figure 7.** The surface pressure distribution on the yttria-stabilized zirconia (YSZ) lattices simulated using the finite element method (FEM). Displayed yellow and blue contours indicate high and low-pressure loads applied from the molten salt, permeating and moving between the sterically propagated 12-coordinated lattices.

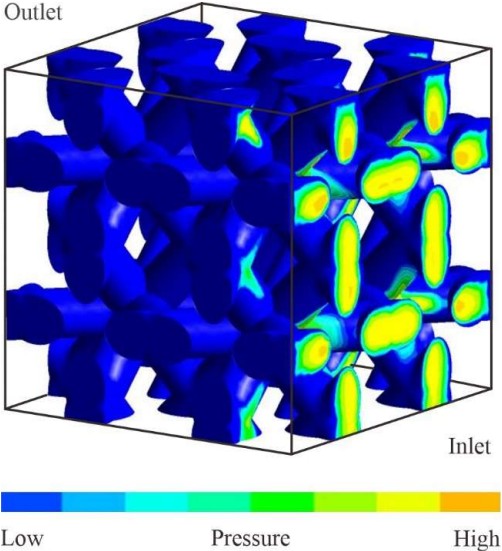

**Figure 8.** The surface temperature distribution on the yttria-stabilized zirconia (YSZ) lattices simulated by the finite element method (FEM). Displayed red and orange contours indicate high and low-temperature distributions transferred for the molten salt flowing and filling the sterically propagated 12-coordinated lattices.

For the mixing and dispersing processes of the YSZ nanoparticles in the acrylic resin, the volume fractions were increased in a stepwise manner from 10, 20, 30, 40, to 50%. The obtained paste viscosities were increased according to the particle volume fractions, and the clayey flowabilities were greater than 40% in the volume fraction. The paste composition was systematically optimized and maximized at a particle volume fraction of 40% to realize smooth spreading during the STL-AM. Kinematic viscosities were measured after the mechanical mixing. The rheological profiles were saturated at above 15 min in the total process time. The kinematic viscosities exhibited similar rheological profiles because the paste material stood for several hours. The optimally prepared paste material ensured stable and continuous operations during the STL-AM.

The numerical variations of the curing depths and part accuracies were measured according to the laser irradiation power, as shown in Figure 9. In the systematic optimizations of the lithographic conditions to create the film specimens, as shown in Figure 9, the curing depth equal to the layer thickness gradually increased according to the irradiation power, and the part accuracies were estimated from the hole diameters via optical scattering by the dispersed nanoparticles. Considering the beam focal diameter of 50 μm as the drawing tolerance for the horizontal directions, the stacking interval used during the STL-AM was defined at 50 μm as the stacking tolerance for vertical directions. Using experimental results from previous investigations, the irradiation power was set to 150 mW, corresponding to a curing depth of 75 μm, and layer laminations with an interval of 50 μm were realized through rigid interlayer bonding [13]. Under the selected conditions, the surplus width reached 65 μm. The laser beam spot was sifted at 65 μm in the inner area of the drawing patterns to offset the surplus width.

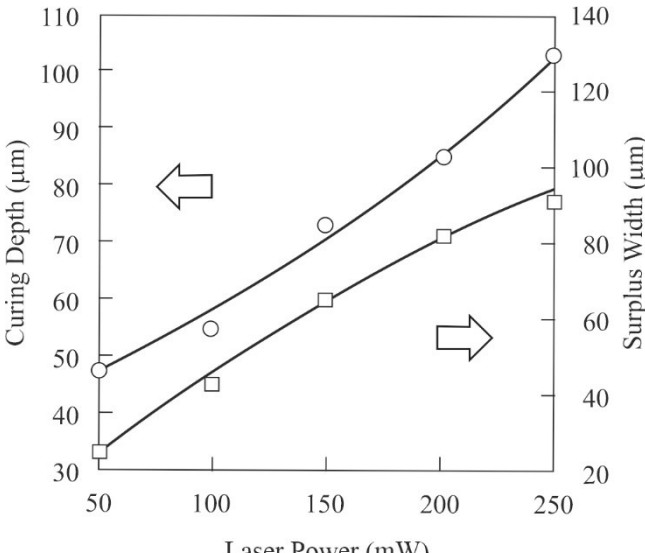

**Figure 9.** The numerical variations of the curing depths and dimensional tolerances according to the irradiation powers in laser drawing. Correlation graphs were obtained from the systematic measurements using the resin pastes with the yttria-stabilized zirconia (YSZ) particles.

The acryl lattice with the YSZ nanoparticles fabricated by STL-AM is shown in Figure 10. In this case, 12 coordination lattices were formed according to the designed model. The sintered lattice of the solid electrolyte was obtained by heat treatments in air. Through the digital measurements of the nodal point distances and lattice diameters in the DOM observations, the linear and volumetric shrinkages after dewaxing and sintering were estimated to be 25% and 60%, respectively. These shrinkage ratios were fed back into the model designs to obtain the intended geometries. The improved part accuracy was eventually achieved at ±10 μm. The cross-section of the YSZ lattice was observed by SEM,

as shown in Figure 11. A fine ceramic structure was obtained without microcracks or pores. The relative density was determined to be 99.5%.

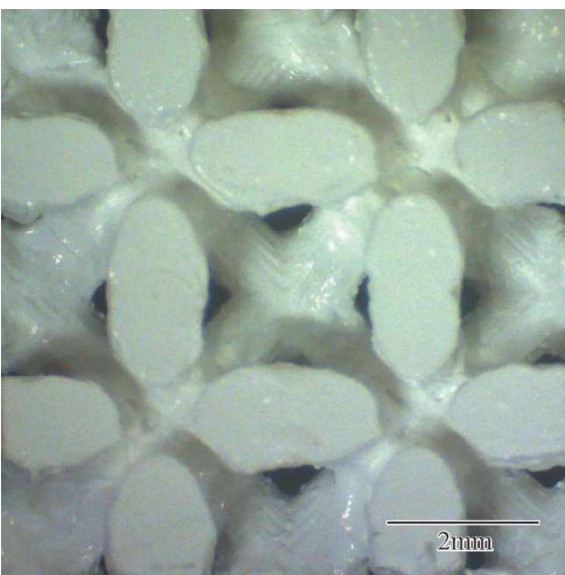

**Figure 10.** The acryl microlattices with the YSZ nanoparticles. The structure will be transferred into a dendritic electrode. The stereolithographic additive manufacturing (STL-AM) components were heat-treated at 1400 °C for 2 h in air. The 12-coordinated lattices were processed at a dimensional accuracy of ±10 μm.

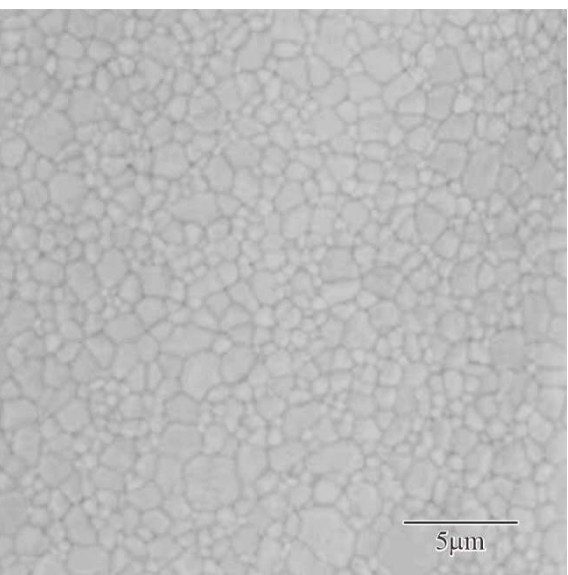

**Figure 11.** Ceramic microstructure of the sintered solid electrolyte in the dendritic microlattices observed by scanning electron microscopy (SEM). The fine crystal grains of the yttria-stabilized zirconia (YSZ) were obtained by systematically optimized heat treatments at 1400 °C for 2 h in air.

In the previous study, YSZ electrodes with planar surfaces were utilized to realize oxygen adsorption in the molten salts [14]. The separation efficiency of oxygen gas was measured as $6.26 \times 10^{-4}$ mol/s·m$^2$. In this study, dendritic lattice patterns were systematically introduced into the YSZ electrode, which effectively in-creased the surface area 3.2 times. Product inspections showed that the oxygen ion conductivity of the used YSZ electrolyte was sufficient at 0.04 S/cm above 800 °C. Therefore, the newly developed system for aluminum refining using ceramic dendrites will be able to continuously produce

oxygen gas in the same molal quantities as carbon dioxide discharges through conventional deoxidization processes using carbon electrodes.

### 4. Conclusions

Dendritic structures of YSZ were successfully processed by STL-AM. The formed YSZ electrodes with ordered porosities were introduced into molten salts to separate oxygen during aluminum smelting. Compared with the conventional process with graphite electrodes, the use of YSZ electrodes is a promising method to reduce carbon dioxide excretion as a greenhouse gas.

In computer graphic modeling, cylindrical microlattices are spatially connected to maximize specific surface areas. Streamline distributions of the molten salts in the lattice patterns were simulated and visualized to realize smooth flowabilities by CFD. Through the computational optimizations, the cylindrical lattice was determined at an aspect ratio of 2.18, and the nodal points were defined at a coordination number of 12. To verify and discuss the durability of ceramic electrodes in molten salts, the distribution profiles of temperature and pressure were recorded and displayed on the lattice surfaces using the FEM.

As the STL-AM material, the YSZ particles were dispersed into photosensitive acrylic resin at a maximum volume fraction of 40%. A UV laser with a beam diameter of 50 μm was scanned on the smoothly spread paste to precisely create cross-sections, and the formed layers were laminated to fabricate solid components. The UV laser irradiation power was optimized at 150 mW to realize strict interlayer bonding, and the inner area of the drawing profiles was sifted at 65 μm to offset superfluous curing by light scattering. The composite precursors were dewaxed and sintered, and the shrinkage ratios were fed back into the model designs. The dendritic YSZ electrode with 99.5% relative density exhibited a part accuracy of $\pm10$ μm.

**Author Contributions:** Conceptualization, S.K.; methodology, M.T. and S.K.; validation, M.T. and S.K.; formal analysis, M.T.; investigation, M.T.; resources, M.T.; data curation, M.T.; writing—original draft preparation, M.T.; writing—review and editing, S.K.; visualization, M.T. and S.K.; supervision, S.K.; project administration, M.T. and S.K. All authors have read and agreed to the published version of the manuscript.

**Funding:** This research received no external funding.

**Institutional Review Board Statement:** Not applicable.

**Informed Consent Statement:** Not applicable.

**Data Availability Statement:** Not applicable.

**Conflicts of Interest:** The authors declare no conflict of interest.

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
