# Peer review of "Stereolithographic Additive Manufacturing of Zirconia Electrodes with Dendritic Patterns for Aluminum Smelting"

_applsci, doi:10.3390/app11178168_

Round 1

Reviewer 1 Report

This manuscript describes the stereolithographic additive manufacturing of zirconia electrodes with dendritic patterns for aluminium smelting. This is an interesting work in which the authors manage to reduce carbon dioxide excretion during aluminium smelting by using YSZ electrodes. Additionally, the YSZ particles were dispersed into photosensitive acrylic resin, so that a UV laser could be used to create cross-sections, which layers were laminated to fabricate solid components. The paper is very well written and the featured results are undoubtedly promising. The document contains high quality figures and diagrams and all the fabrication process of the structures is properly described, for the sake of reproducible research. The Conclusions are perfectly supported by the results and the latter state a dendritic YSZ electrode with 99.5% relative density and a part accuracy of +-10um, which is definitely great. All in all, this is a valuable work and an excellent match for Applied Sciences. I consider that it should be accepted for publication in its present form, not having any additional suggestions or comments to add.

Author Response

Thank you very much for reviewing my paper.
This paper has been checked by a native English speaker, so the English style and spelling are considered correct.    Best regards, Masaya Takahashi

Reviewer 2 Report

Zirconia electrodes with dendritic patterns were fabricated by stereolithographic additive manufacturing. The modeling and optimization techniques were used and YSZ electrodes were introduced as a promising method to reduce carbon dioxide. 

The manuscript is well-composed, written, and nicely discussed. In my opinion, the manuscript is suitable enough for publication in Applied Sciences 

Author Response

(The authors gave the same response as above.)

Reviewer 3 Report

This paper describes the design and preparation of Zirconia electrodes with dendritic patterns by STL-AM for aluminum smelting. The methodologies used for the design and STL-AM seems appropriate and the electrodes were apparently synthesized. However, we need to observe the performance of these novel electrodes in aluminum smelting and the authors should present some results obtained with this new system and comparing them with others.

Author Response

Dear Reviewer,

Thank you very much for your kind revising my paper. My manuscript has been revised according to your suggestion. I would like to answer for your comments as follows.

Best regards,

Masaya Takahashi

Osaka University, Japan

Comment: This paper describes the design and preparation of Zirconia electrodes with dendritic patterns by STL-AM for aluminum smelting. The methodologies used for the design and STL-AM seems appropriate and the electrodes were apparently synthesized. However, we need to observe the performance of these novel electrodes in aluminum smelting and the authors should present some results obtained with this new system and comparing them with others.

Answer: According to your suggestion, I have added the following paragraph as the Line: 244 - 252 with a reference [14] into the last part of 3. Results and Discussion to compare the new and conventional systems for aluminum refining.

“In the previous study, YSZ electrodes with planar surfaces were utilized to realize oxygen adsorption in the molten salts [14]. The separation efficiency of oxygen gas was measured as 6.26×10-4 mol/s‧m2. In this study, dendritic lattice patterns were systematically introduced into the YSZ electrode, which effectively increased the surface area 3.2 times. Product inspections showed that the oxygen ion conductivity of the used YSZ electrolyte was sufficient at 0.04 S/cm above 800 ºC. Therefore, the newly developed system for aluminum refining using ceramic dendrites will be able to continuously produce oxygen gas in the same molal quantities as carbon dioxide discharges through conventional deoxidization processes using carbon electrodes.”

Round 2

Reviewer 3 Report

The authors have answered satisfactorily to the referees questions.